# Micro Spectrometers Based on Materials Nanoarchitectonics

**DOI:** 10.3390/ma16062253

**Published:** 2023-03-10

**Authors:** Yanyan Qiu, Xingting Zhou, Xin Tang, Qun Hao, Menglu Chen

**Affiliations:** 1School of Optics and Photonics, Beijing Institute of Technology, Beijing 100081, China; 2Yangtze Delta Region Academy, Beijing Institute of Technology, Jiaxing 314019, China

**Keywords:** micro spectrometers, nanomaterials, nanoarchitectonics, spectral analysis

## Abstract

Spectral analysis is an important tool that is widely used in scientific research and industry. Although the performance of benchtop spectrometers is very high, miniaturization and portability are more important indicators in some applications, such as on-site detection and real-time monitoring. Since the 1990s, micro spectrometers have emerged and developed. Meanwhile, with the development of nanotechnology, nanomaterials have been applied in the design of various micro spectrometers in recent years, further reducing the size of the spectrometers. In this paper, we review the research progress of micro spectrometers based on nanomaterials. We also discuss the main limitations and perspectives on micro spectrometers.

## 1. Introduction

Elements and their compounds known on Earth show specific spectra; these spectra are thus regarded as the fingerprints that identify the elements themselves. Due to this, the composition, structures, and physical states of substances can be determined by spectral analysis. Spectral analysis has a number of advantages, such as being fast, accurate and non-destructive, leading it to play a remarkable role in substance identification, physicochemical property analysis and content detection. To resolve such spectra, the incident light must be analyzed by optical spectrometers, which decompose composite light mixed with different wavelengths into spectral curves; then, the information regarding wavelengths and intensity is obtained. There are various types of optical spectrometers; in addition to those used in the visible band, there are also ultraviolet and infrared types. Today, spectrometers have a wide range of applications in biosensing [1], industrial inspection [2], mineral exploration [3], environmental monitoring [4], chemical analysis [5], deep space exploration, military technology, etc.

Traditional high-performance benchtop spectrometers, which can provide ultrahigh resolution and a wide spectral range, generally consist of three parts: namely, discrete dispersion elements, mechanical movable parts and detector arrays [6,7,8]. However, because of this complex structure, spectrometers usually possess many problems such as large volume, high energy consumption and high cost. For some emerging applications, the need for spectrometer miniaturization and integration is much greater than the requirement for high performance. As a result, the development of micro spectrometers has naturally become an important research direction of spectral instruments. With the development of micro nanotechnologies, computer technologies and other fields, micro spectrometers have emerged in recent years [9,10]. Micro spectrometers not only have a low cost, low power consumption and convenience in field detection, but they can also be redeveloped, so the application fields are greatly expanded.

Since the early 1990s, researchers have presented micro spectrometers via various designs and working principles. In the early stage of micro spectrometer development, the main focus lay in system integration. In 1990, D. Goldman et al. first studied and prepared the integrated waveguide miniature spectrometer based on the integrated splitter device combining planar waveguide and grating [11]. Afterwards, researchers began paying attention to shrinking the system components and system structure. In 1992, Dr. Mike, an American scientist and founder of the Marine company, successfully developed the world’s first micro-optical fiber spectrometer, namely the micro-optical fiber spectrometer S1000. This micro spectrometer was compact: one-tenth the size of a conventional spectrometer. As a result, production costs were greatly reduced, and great progress was made in the portability of the spectrometer. Since then, micro spectrometers have been developed [12,13,14,15,16] and gradually used in industry. However, in some applications, the spectrometer needs to be further shrunk. Yet, reducing the size of the optical and detection elements leads to a significant decrease in the spectral resolution, sensitivity and dynamic detection range of the spectrometer. Thus, traditional scale methods are facing technical challenges.

The development of nanomaterials [17,18,19,20,21,22] provides researchers with a new method of thinking to shrink the spectrometer. Nanomaterials, such as zero-dimensional nanomaterials (quantum dots), one-dimensional nanomaterials (nanowires or nanorods) and two-dimensional nanomaterials (graphene, black phosphorus, etc.) have low cost, high carrier mobility, adjustable bandgaps and a precisely controlled materials growth process. In particular, nanomaterials exhibit many novel optical and electrical properties due to the quantum confinement effect; therefore, they show attractive prospects in electrical and optical devices and have attracted wide attention and research all over the world. Thus, it is of great significance to use this new nanomaterial for spectral filters or the photodetector units of spectrometers. In recent years, researchers have designed a variety of spectrometers by using nanomaterials, some of which do not even require optical components, allowing the spectrometers to be further miniaturized.

In this article, we review the recent relevant results on micro spectrometers based on nanomaterials and nanoarchitectonics. Depending on the dimension of nanomaterials used for the spectrometer’s filter or detector units, the micro spectrometers can be divided into three categories: zero-dimensional nanomaterials-based micro spectrometers, one-dimensional nanomaterials-based micro spectrometers and two-dimensional nanomaterials-based micro spectrometers. The progress in micro spectrometers based on materials nanoarchitectonics is summarized in Figure 1.

## 2. Types of Micro Spectrometers

Micro spectrometers constructed by nanomaterials have different structures and operating modes from traditional spectrometers, as well as a series of advantages, such as structure, cost, integration, stability and so on. There have been many reports on micro spectrometers based on nanomaterials at home and abroad. In this review, we divide the type of micro spectrometers based on nanomaterials.

### 2.1. Zero-Dimensional Nanomaterials-Based Micro Spectrometers

The typical zero-dimensional (0D) nanomaterial representative is quantum dots (QDs). QDs have the characteristics of broadband absorption and narrowband emission, and the absorption spectra can be precisely tuned by varying their composition and size. Thus, QDs have fine spectral responses over a wide wavelength range. For example, by tuning the particle size, the optical response of cadmium chalcogenide QDs [23,24,25,26,27] could cover the visible wavelength, while lead [28,29,30,31] or mercury chalcogenide QDs [32,33,34,35,36,37,38,39,40,41] could cover infrared regions to the terahertz region. Due to this property, QDs are excellent for identifying colors or the spectra of matter.

In 1997, Jimenez et al. proposed a sensitive QD spectrometer which was created in the InAs/GaAs/Al_x_Ga_1−x_As material system. This device consisted of two main parts. An inhomogeneous QD plane was used to obtain the spectra information, while a resonant-tunneling device was used to carry out the spectra readout. It could integrate into an array because its size was small, similar to that of a common semiconductor photodetector [42]. In 2015, Zhang et al. designed an embedded micro spectrometer based on a 64-pixel QD photodetector array, showing a good spectral response in the 500–1000 nm band, especially 700–900 nm. This detector was a GaAs n-i-n photodetector with a double-barrier AlAs, which implanted an InAs QDs into its thin quantum well. The spectrometer had high sensitivity and could detect weak light down to 10^−14^ W, which was more suitable for the spectral detection of biological micro-regions [43]. Similarly, in 2018, Wang et al. also made use of InAs QDs and a In_0.15_Ga_0.85_As quantum well to design an AlAs double-barrier structure. The sensitivity of the photodetector based on this structure was significantly increased. They combined this 64-pixel QD photodetector with a readout circuit to create an android-based micro spectrometer. The QD micro spectrometer consisted of two main structures, namely the optical platform and the optoelectronic system. It had been experimentally proven that the proposed spectrometer was more sensitive than a commercial charge-coupled device (CCD) spectrometer. Moreover, this spectrometer had promising applications in the detection of micro-regions in biological samples [44].

In addition, colloidal QDs (CQDs) can be synthesized by solution treatment and be processed, molded and integrated with a liquid state, which will greatly miniaturize spectrometers at a low cost. Hence, CQDs are an attractive nanomaterial for micro spectrometers.

In 2015, Bao et al. demonstrated a micro QD spectrometer, which used CQDs as a spectral filter array for the first time and integrated it with a CCD. They chose 195 spectra-tunable cadmium sulfide (CdS) and cadmium selenide (CdSe) CQDs materials, which were obtained by adjusting the size or composition of CQDs and had different wavelength-selective transmittance in the selected spectral range. The CQD filters and the integrated QDs spectrometer are shown in Figure 2a. The spectral measurement of this micro spectrometer was based on the principle of wave-division multiplexing, computationally reconstructing the original spectrum by measuring the total transmission intensity of each given CQD filter. A spectral resolution of ~3.2 nm was achieved in the spectra range of 390–690 nm [45]. This work successfully demonstrates that CQDs are ideal broadband filter materials for use in micro spectrometers.

In 2019, Tang et al. took advantage of the excellent optical properties of HgTe CQDs to design a CQD hyperspectral sensor, which could acquire hyperspectral image cubes in the short-wave infrared range. The device consisted of HgTe CQD photovoltaic sensors and distributed Bragg reflector filter arrays. The spectral measurement of this hyperspectral sensor showed a spectral resolution of up to 180 [50]. In 2020, Zhu et al. reported a broadband perovskite QD (PQD) spectrometer with a spectra bandwidth of 750 nm (250–1000 nm) and a spectral resolution of ~1.6 nm, and both parameters were superior to that of human visualization. In this study, 361 non-emissive in situ fabricated PQD-embedded films (PQDFs) were selected to form a filter array with which a silicon-based photodetector array was integrated. Further, a total-variation optimization algorithm based on compressive sensing was used to reconstruct the spectral information. The architecture of the developed PQDF-based hyperspectrometer is shown in Figure 2b [46]. To address the problem of a lack of high-resolution NIR (near-infrared) QD filters, in 2021, Li et al. proposed a NIR QD spectrometer. They combined 195 PbS and PbSe CQD filters with a NIR CMOS (complementary metal oxide semiconductor) detector and utilized the compressive sensing-based total-variation algorithm, achieving a spectra resolution of 6 nm in the NIR wavelength range between 900 and 1700 nm [47]. The schematic diagram of the proposed NIR QD spectrometer is shown in Figure 2c.

In 2021, Venettacci et al. fabricated an algorithm-based spectrometer covering the whole visible-NIR range by utilizing narrowband PbS CQDs. They drop-coated the PbS CQD dispersions onto a planar glass substrate as a filter and then placed it over a photoconductive detector whose active layer material was also PbS CQDs, thus forming a spectra-selective detector. They prepared seven batches of PbS CQD dispersions and combined the filters with photodetectors, eventually obtaining 28 devices with different responsivity spectra. The schematic diagram of the proposed spectrometer is shown in Figure 2d. For the spectral reconstruction and the peak wavelength identification spectral reconstruction, they achieved a maximum resolution of 40 nm and a mean error of 5 nm [48]. In June 2022, a single-dot spectrometer based on the 5% LiCl-doped perovskite film was proposed by Guo et al. For the structure of this single-dot spectrometer, the top electrode was a silver probe, the active layer was the perovskite film (~500 nm thickness), and the materials located above and below it were the Spiro-OMeTAD layer and the SnO2 layer, respectively. A 5 nm spectral resolution was achieved in the working area footprint of 440 × 440 µm^2^. Moreover, this spectrometer showed a good linear response in the irradiance range of 0.42 µW cm^−2^ to 120 mW cm^−2^. They showed an on-chip integrated spectral imaging system and obtained spectral information by spectral reconstruction [51].

The interferometric platforms of Fourier transform spectrometers have been shrunk in recent years, but their signal detection still relies on external imaging sensors [52,53]. To further address the challenge of the sensor miniaturization of waveguide spectrometers, in October 2022, Grotevent et al. integrated a piece of subwavelength HgTe CQD photodetector into a LiNbO_3_ waveguide to demonstrate an ultra-compact (below 100 μm × 100 μm × 100 μm) miniature Fourier transform waveguide spectrometer (Figure 2e). This short-wave infrared spectrometer, operating at room temperature, had a spectral response at a maximum wavelength of 2 μm and a moderate spectral resolution of 50 cm^−1^ [49]. The design of this ultra-compact spectrometer offers a new idea for the miniaturization of the Fourier transform spectrometer, which is expected to be applied to hyperspectral cameras, consumer electronics, etc., in the future.

Overall, the QDs spectrometer represents a great breakthrough in the miniaturization of spectrometers. QDs are highly tunable, tiny and light-sensitive semiconductor crystals. In theory, the use of QDs can miniaturize the spectrometer without affecting its spectral resolution, spectral range and efficiency. However, micro spectrometers based on more environmentally friendly CQD materials, such as ZnSe and InP, should be developed. In the future, such tiny QDs spectrometers may be used in personalized medicine, microfluidic chip laboratories, fluid detection [54] and other fields.

### 2.2. One-Dimensional Nanomaterials-Based Micro Spectrometers

One-dimensional (1D) nanomaterials, such as nanowires and nanotubes, have shown attractive prospects in electrical and optical devices [55,56,57,58] in recent years. Since the nanowire and nanotube diameter reduces in size, singularities in the electronic density of states develop at special energies, called van Hove singularities. The electronic density of states is large and bears closer resemblance to molecules and atoms, but sharply differs from crystalline solids. Thus, within the limits of small diameters, novel quantum phenomena associated with this characteristic 1D density of state features are observed. Moreover, with the development of nanotechnology, the growth technology for nanowires and nanotubes has been advanced to a level in which the required composition, structure and heterojunction can be easily synthesized. As a result, the difficulty of micro spectrometer integration can be greatly reduced. Using 1D nanomaterials can even combine the spectral splitting unit and detection unit, which will greatly reduce the volume of the spectrometer. In recent years, there has been much research on 1D nanomaterials-based spectrometers.

In 2011, Cavalier et al. integrated a SiN single-mode ribbed waveguide interferometer with an array of 24 superconducting epi-NbN nanowire single-photon detectors, exhibiting an infrared SWIFTS (Stationary Wave Integrated Fourier Transform Spectrometer) device. The nanowire arrays they designed had a width of 40 nm, a thickness of 4 nm and a space of 120 nm. After a colored light of around a 1.55 μm wavelength was introduced to the optical waveguide, the nanowire arrays would produce a counter-propagative stationary interferogram. Using the SNSPD (Superconducting Nanowire Single-Photon Detectors) operating in single-photon counting mode, the modulation of the source bandwidth was detected. With 4.2 K optical power modulation, a spectral resolution of 170 nm and a spectral width of 2 μm were achieved. The operating principles of the SWIFTS-SNSPD device are shown in Figure 3a [59]. In 2017, French et al. demonstrated a hyperspectral imaging spectrometer making use of 1.7 µm thin-layered gallium phosphide (GaP) nanowires as a multiply scattering medium in combination with a compressive sensing approach. This scattering medium was only a few microns thick and was characterized by uniformity and high dispersion. In the illuminated area, there were 100 separate transmission channels in the nanowire mats, which was sufficient for spectral information encoding. When light passed through the nanowire mats’ scattering medium, a series of speckled patterns appeared, which were then imaged onto the camera [60]. In June 2019, Uulu et al. first presented a Fourier transform plasmon resonance (FTPR) nanospectrometer. In this work, a metal nanowire or ridge was deposited beside the nanoslit, tilting five degrees to the nanoslit principal axis (Figure 3b). In this way, the SPPs excited by the Pt nanowire at the interface between the metal and the dielectric would combine with the incident light whose polarization was p-polarized, thus generating the interference fringes of SPP waves. After the transmitted SPPs decoupled from the metal surface, they would be collected by an objective in the far field and finally detected by a CCD sensor. By applying fast Fourier transform (FFT) to the fringe pattern of the SPPs, the spectral information of the incoming light was acquired. This nanospectrometer, based on the interference of the plasmon wave in the subwavelength metal nanoslit, can be used to sense the refractive variation and measure plasmon–exciton hybridization in a small region [61].

In September 2019, Cheng et al. reported their efforts in the first broadband on-chip single-photon spectrometer. They used a continuous NbN nanowire with a length of 7 mm to function as a multi-channel spectral-resolving single-photon detector, which was integrated with an on-chip dispersive echelle grating (Figure 3c). The spectrometer covered visible and infrared bands from 600 nm to 2000 nm, providing more than 200 equivalent wavelength detection channels with further scalability [62]. In the same month, Yang et al. used special CdS_x_Se_1−x_ nanowires with a graded bandgap to develop a micro spectrometer with a size of only tens of microns. In their work, they used a process similar to that used to make computer chips to fabricate an array of photodetectors on the nanowires. Since each detector had varying responses to different colors of light, the team reconstructed the spectral information that needed to be measured from the equations of response functions by solving the inverse problem. The operational schematic of the nanowire spectrometer is shown in Figure 3d. The photosensitivity of the photodetector units reached 1.4 × 10^4^ AW^−1^, with a response speed of ~1.5 ms and a recovery time of ~3.5 ms. With further development, this spectrometer may be used for disease surveillance and food safety testing [63].

Instead of varying composition in a precise manner with the axial position along the nanowire, as in Yang’s work [63], in December 2019, Meng et al. achieved wavelength selectivity by controlling the nanowire radius. They reported a micro spectrometer based on structurally colored nanowires for the first time. This micro spectrometer consisted of an array of vertical silicon nanowire (Si NW) (doped p+/i(n−)/n+) photodetectors formed above an array of planar photodetectors (doped n+/i(n−)/p+), covering the entire visible spectrum from 400–800 nm. The NWs with wavelength selectivity could be regarded as acting as a filter for the planar photodetector. They utilized the recursive least squares algorithm (for broadband spectra) and lasso regression (for narrowband spectra) to reconstruct the incident spectra [68].

Arrayed waveguide gratings compatible with waveguide-integrated single-photon detectors have limitations in bandwidth, thus only providing a limited number of spectral channels. Therefore, to enable high-resolution broadband optical operation, in March 2020, Hartmann et al. showed a fully integrated on-chip spectrometer based on tailored disorder for broadband light scattering. The device, fabricated on a silicon nitride platform (Si_3_N_4_), contained fiber-to-chip couplers, the nanophotonic structure and single-photon detectors. The light was launched by an input waveguide into a semicircular scattering region with a radius of 100 μm and randomly distributed air pores with a radius of 125 nm. The light diffused and was coupled into 16 output waveguides; eventually, it was detected by 16 SNSPDs, respectively (Figure 3e). Spectral-to-spatial mapping by the transmission matrix at the system using the SNSPDs allowed the reconstruction of a given signal. The resolution of this spectrometer reached 4 nm at NIR wavelengths and was improved to 30 pm at shorter wavelengths. Moreover, such a spectrometer had high sensitivity and was able to reconstruct the detection signal even at a very low input power of −111.5 dBm [64].

In April 2020, Zheng and co-workers, presented a monolithic spectrometer with a dispersive photodiode array integrated into a single composition-graded CdS_x_Se_1−x_ nanowire (Figure 3f). The device operated in a waveguide mode. The nanowire, enhancing light–matter interaction, was the CdS of a 2.42 eV (optical) bandgap on one end and the CdSe of a 1.74 eV bandgap on the other, and it functioned as a wavelength selective component. A 5 nm spectral resolution and 10^13^ Jones room-temperature detectivity were exhibited. This presentation has made new progress in high-resolution and sensitive detection in on-chip spectroscopy [65]. In July 2020, Yun et al. demonstrated a design for a single-photon spectrometer using SNSPDs in cascaded photonic crystal (PC) structures (Figure 3g). The structure of this spectrometer contained a wavelength division multiplexing filter with the ability to resolve wavelengths, which was composed of a bus waveguide coupled to a planar array of PC nanocavities with different resonant wavelengths. Since the superconducting nanowires were located next to a single cavity, there was an evanescent coupling between the two, which could allow the nanowires to absorb the light captured in the cavity, and thus, detect the presence of a single photon. Through their analysis, absorption efficiencies reached ~80% in the structure with four cascaded cavities or two cascaded cavities, and the spectral resolution achieved was about 1 nm [66].

Designing a spectrometer that does not require wavelength-multiplexed optics can effectively reduce complexity and physical footprint. In 2021, Wu’s research group from Nanjing University reported an optics-free single-detector spectrometer using a superconducting nanowire. They used the nonlinear spectral modulation properties of an SNSPD and computational spectral algorithms to realize spectral detection. This micro spectrometer had a wide spectral response capability covering 660 nm–1900 nm and realized a spectral resolution of sub-10 nm at the telecom. The operating principles diagram of the spectrometer is demonstrated in Figure 3h. They also combined a LiDAR with this spectrometer to demonstrate the multi-spectral LiDAR capabilities [67].

An SNSPD is suitable for detecting the spectrum of weak light. In 2023, Zheng et al. reported their efforts on a photon-counting reconstructive spectrometer that was formed on a silicon-on-insulator (SOI) substrate with a silicon layer thickness of 340 nm. The team fabricated the SNSPDs on metasurfaces with different structural parameters; then, the spectral response of the SNSPDs was modulated according to the corresponding metasurfaces. Combined with the compressive sensing algorithm, the spectrometer achieved spectral reconstruction of monochromatic light in the wavelength range of 1500–1600 nm with a spectral resolution of 2 nm. At the same time, the measurement time in this band was greatly reduced, and the detection efficiency was 1.4–3.2% [69].

One-dimensional nanomaterials, especially nanowires, have been gradually used in micro spectrometers because of their excellent light absorption ability and support for simple line array construction. As can be seen from the reports summarized above, some nanowire spectrometers no longer require optical components, which greatly reduces the size of the spectrometer. In the future, such miniature spectrometers are expected to be used in applications such as electronic consumer and wearable electronic devices. However, for use in practical production, more research work is still needed to control the size and growth of high-quality 1D nanomaterials and to address the large-area integrated positioning problem of 1D nanomaterials.

### 2.3. Two-Dimensional Nanomaterials-Based Micro Spectrometers

Since the success of the mechanical stripping of graphene in 2004, the exploration and research of 2D materials have developed rapidly. Two-dimensional nanomaterials have been at the forefront of research due to their thickness at the atomic scale and their unique optoelectronic properties. In recent years, 2D nanomaterials, such as hexagonal boron nitride (hBN), transition group metal sulfides, and black phosphorus, (BP) have been discovered and developed rapidly, greatly expanding the scope of applications of 2D materials. They show many novel applications and bright prospects in the field of photoelectric detection [70,71,72,73,74]. The optical detection of 2D materials has advantages due to their strong light–matter interactions, sharp atomic interfaces and electrically tunable optical responses. For example, BP [71,75,76,77] is a 2D semiconductor material with a controllable band gap and high carrier mobility. The band gap (0.3–1.5 eV) of thin-layer BP can be adjusted by the thickness. Monolayer BP has a wide photoresponse range (1–5 μm) and can strongly interact with electromagnetic waves. Therefore, 2D materials are demonstrated as an intriguing material for photodetection and have great application value in spectrometers.

In 2016, Wang et al. designed a graphene-based spectrometer operating in the mid-infrared wavelength range. The proposed spectrometer used Fabry–Pérot cavities as filters. The transmission wavelength of the spectrometer at room temperature varied with the voltage applied to this cavity. Meanwhile, graphene acted as a surface waveguide, which could spread the SPPs excitated by periodic grating structures into the Fabry–Pérot cavities (Figure 4a). This structure had a micron-sized footprint and was compatible with CMOS processing [78]. In 2018, Liu et al. showed a micro spectrometer with a plasmon-tunable graphene photodetector (Figure 4b). This photodetector, based on a graphene/monolayer MoS_2_ vertical heterostructure, had a photoresponsivity of up to 10^7^ AW^−1^ at room temperature. While the device had merged mini-filters and detectors needed in a spectrometer, it could still be arranged as a two-dimensional array with different bias voltages to detect spectral information in the spectrum range of 6–16 μm. In this way, more spectral information would be collected, which was convenient to realize spectral reconstruction [79].

Taking advantage of the tunable energy band of BP in the mid-infrared band, in 2021, Yuan et al. produced a BP spectrometer in the 2–9 µm spectral range. This spectrometer was based on a single tunable BP photodetector (Figure 4c); thus, its active area footprint was very compact at only 9 × 16 µm^2^. The implementation concept of the spectrometer was based on the strong photoresponse and Stark effect of the BP photodetectors. There was an hBN/BP/hBN heterostructure in this device’s structure, which was capped by a monolayer of graphene. The source-drain electrodes consisted of chromium/gold, while the silicon substrate and the mid-infrared transparent graphene operated as the back and top gates, respectively. This compact single-detector spectrometer had medium spectral resolution and was more suitable for sensing applications of spectral information [80]. In June 2022, Zheng et al. used a dual-gate BP phototransistor to create a BP monolithic spectrometer. This single-detector spectrometer contained a 10 nm BP channel, and the top and bottom were encapsulated by two hBN sheets (roughly 10–20 nm in size). Meanwhile, few-layer graphene sheets (5 nm) functioned as the source, drain and top gate electrode of the phototransistor (Figure 4d). This single-detector spectrometer operated in the broadband mid-infrared range with the ultra-fine spectral and temporal resolution, which were ~2 cm^−1^ and 2 ms, respectively [81].

Van der Waals junctions (vdW) are artificial materials formed by stacking two-dimensional materials in a specific order, which have excellent optoelectronic properties. This heterojunction provides highly tunable functions that can solve the problem of an inadequate energy band structure modulation of a single 2D material, therefore allowing the realization of high-resolution broadband spectral sensing. In July 2022, Deng et al. designed a near-infrared micro spectrometer using a 2D van der Waals heterostructure (2D-vdWH). They chose ReS_2_ and WSe_2_ materials and introduced Au atoms into this 2D heterostructure to form a ReS_2_/Au/WSe_2_ sandwich structure. The heavy metal Au atoms enhanced the excited state transition dipole moments between the two-dimensional heterogeneous layer; therefore, the interlayer coupling was significantly enhanced. This structure allowed the spectrometer to achieve an electrically tunable infrared photoresponse in the wavelength range from 1.15 to 1.47 µm. Combined with the regression algorithm, this spectrometer, with an active area of only 6 µm, could accomplish spectral reconstruction and spectral imaging. The schematic drawing of the 2D-vdWH spectrometer is demonstrated in Figure 4e. However, this spectrometer has limited resolution and operates at cryogenic conditions [82]. Later, in October 2022, Yoon et al. demonstrated a micro spectrometer with a single vdW junction that could operate under ambient conditions (Figure 4f). In their work, a MoS_2_/WSe_2_ heterojunction was chosen, and to insulate and passivate it, they encapsulated it with hBN layers on the top and bottom. This high-performance spectrometer could distinguish the peak wavelength of monochromatic light and resolve the broadband spectrum, with an accuracy and resolution of 0.36 nm and 3 nm, respectively. The proposed single-junction spectrometer, operating in the wavelength range of ~405–845 nm, was compact; it had an active area of ∼22 mm by 8 mm and was able to provide scalability. Its concept could be extended to other junctions and had the advantage of compatibility with CMOS and photonic integrated circuits. It was expected to have a broad application prospect in smartphones, satellite devices, etc. [83].

Two-dimensional nanomaterials possess excellent optical detection characteristics. As seen in the above demonstrations, several types of 2D nanomaterials have been successfully used in micro spectrometers. Moreover, most of these spectrometers have a micron footprint. However, due to the ultra-thin thickness of 2D nanomaterials, their light absorption performance may be poor, so further work is needed to optimize device design and manufacturing techniques. Moreover, there is an urgent need for a micro-spectrometer via 2D nanomaterials with a good light response, wide band and high detection sensitivity. Therefore, micro spectrometers via 2D nanomaterials will face many challenges in the future.

In conclusion, over the past three decades, spectrometers have been moving toward miniaturization. Especially, researchers have designed a variety of spectrometer structures by utilizing nanomaterials and combined them with computational reconstruction algorithms to develop many novel micro spectrometers. The performance and structures of micro spectrometers based on materials nanoarchitectonics discussed above are shown in Table 1.

## 3. Conclusions

Using nanomaterials in miniature spectrometers has greatly reduced the cost and is successfully moving spectroscopic technology out of the laboratory and into daily applications—for example, for disease detection and food detection. Nevertheless, there are still some problems to be considered:The detection wavelengths of nanomaterials-based micro spectrometers are mainly concentrated in the visible and near-infrared spectral ranges. Although some of the reported micro spectrometers reach the infrared range, their spectral resolution is relatively low and far below the human visual resolution. In addition, the operating band of most micro spectrometers also needs to be broadened.As the detector and the photon collection area are further reduced, the sensitivity of each instrument design is an increasingly important factor. Therefore, the sensitivity of the detector will need to be higher.Many of the micro spectrometers mentioned above operate at cryogenic conditions, and thus, have limited operations.

According to the above, there is still room to improve the nanomaterials-based micro spectrometer. A wide band, high resolution and high sensitivity are important for micro spectrometers, and for their future development, focus must lie in the following aspects:Broaden the spectral detection range of nanomaterials-based micro spectrometers. Micro spectrometers that respond to multiple wavelength bands may offer great flexibility and enable the development of various spectroscopic applications. In addition, key parameters such as spectral resolution, responsiveness and efficiency also need to be considered.Improve the detection sensitivity of micro spectrometers. For applications requiring low light-level detection, the sensitivity of the spectrometer is an increasingly important factor. For example, improving the signal-to-noise ratio is an effective way to achieve this.Focus on developing micro spectrometers operating at ambient conditions. The operation of the spectrometer needs to be carried out in certain conditions; if these conditions are not met, the instrument will not be used effectively. Operating in ambient conditions can increase the operational flexibility of the spectrometer.Some of the micro spectrometers based on nanomaterials use toxic nanomaterials, such as cadmium and mercury. Although the performance of such materials is high, instruments based on such materials cannot be applied in biological applications and cannot be mass-produced. Therefore, the use of environmentally friendly nanomaterials for micro spectrometers is an important trend.

To summarize, micro spectrometers are developing towards a broadband range, high resolution, high signal-to-noise ratio, high integration, low cost, rapid detection and other directions. Researchers have geared their persistent exploration and efforts towards new principles, new technology and new materials. Nanomaterials have potential applications in spectrometers due to their excellent optical properties. In the future, there may be a strong trend for nanomaterials to replace traditional materials in the field of micro spectrometers. By utilizing nanomaterials, optical component-free micro spectrometers based on photodetectors will become an important development direction.

## Figures and Tables

**Figure 1 materials-16-02253-f001:**
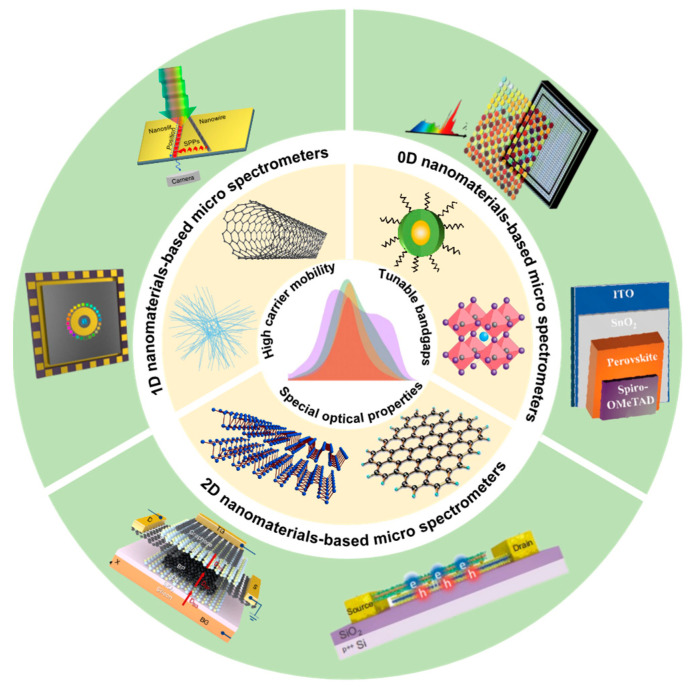
Progress in micro spectrometers based on nanomaterials. The image in the center presents the spectrum. The light-gold annulus around it shows the representative nanomaterials. The green annulus at the outermost area demonstrates some representative structures of these micro spectrometers based on nanomaterials.

**Figure 2 materials-16-02253-f002:**
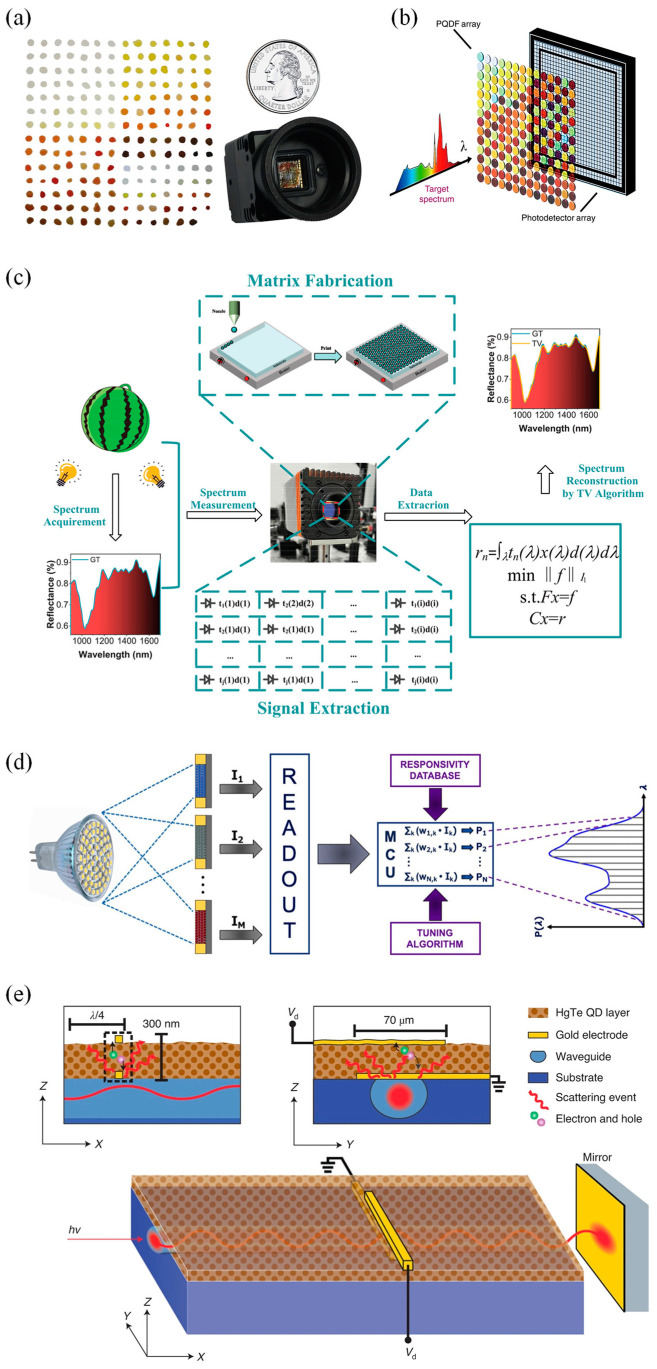
Architecture and schematic diagram of 0D nanomaterials-based micro spectrometers. (**a**) CQD filters and an integrated quantum dot spectrometer. On the left are CQD filters. A total of 195 CQD materials in the form of filters. On the right is a QD micro spectrometer made from the 195 CQD filters and a CCD array detector, which is in the form of a digital camera with electronics and circuits, comparable in size to a US quarter [45]. Copyright 2015, *Nature*. (**b**) The architecture of the proposed PQDF-based hyperspectrometer [46]. Copyright 2020, *Light Science & Applications*. (**c**) Schematic diagram of the developed NIR QD spectrometer, showing the whole process including the acquirement of a real sample’s spectrum, construction of a NIR QD spectrometer, measurement of the spectrum, extraction of the signal and reconstruction of the spectrum by compressive sensingbased total-variation algorithm [47]. Copyright 2021, *Advanced Optical Materials*. (**d**) Schematic diagram of the proposed algorithm-based spectrometer [48]. Copyright 2021, *Photonics and Nanostructures–Fundamentals and Applications*. (**e**) Schematic of the developed waveguide spectrometer. Waveguide spectrometer with a monolithically integrated photoconductor and the respective cross sections. The subwavelength photodetector, created on top of a buried and leaky optical waveguide, contains one bottom gold electrode operating as a scattering center, a photoactive HgTe CQD layer and a top gold electrode [49]. Copyright 2022, *Nature Photonics*.

**Figure 3 materials-16-02253-f003:**
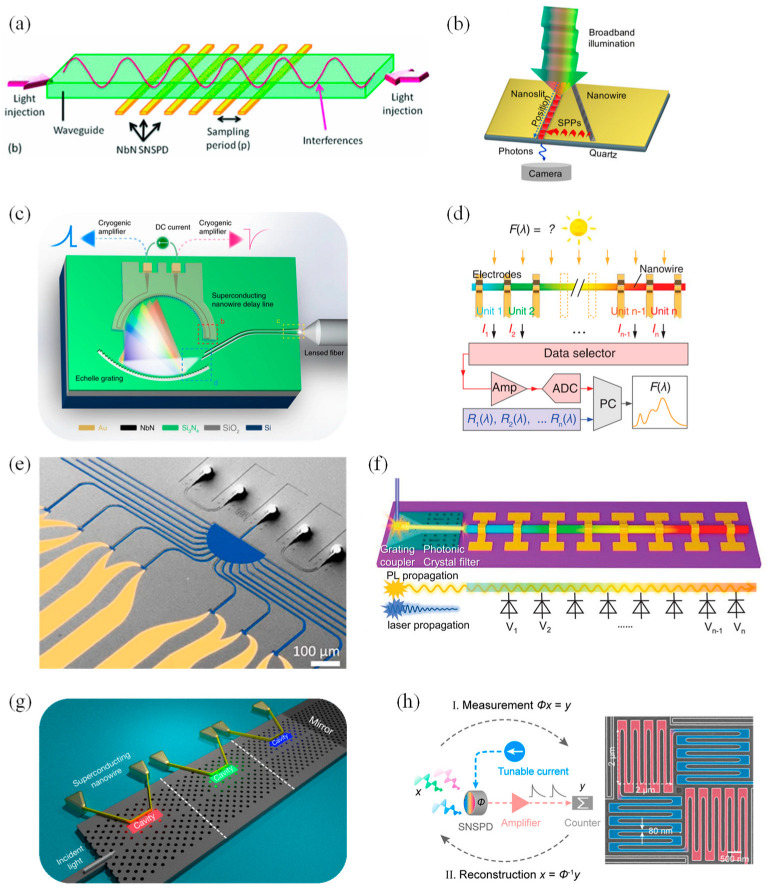
Architecture and schematic diagram of 1D nanomaterials-based micro spectrometers. (**a**) Operating principles of the SWIFTS−SNSPD device. An on−chip−located SNSPD detects a part of the light localized above, within the waveguide [59]. Copyright 2011, *AIP Advances*. (**b**) Schematic drawing of the FTPR nanospectrometer composed of subwavelength slit-nanowire plasmonic interferometer fabricated by focused ion beam (FIB) etching and metal nanowire (or ridge) deposition [61]. Copyright 2019, *Applied Physics Letters*. (**c**) The schematic sketch of the broadband spectrometer. The on-chip focusing echelle grating is used as a wavelength-discriminating component, while the superconducting nanowire acts simultaneously as a single-photon detector and a slow microwave delay line to continuously map the dispersed photons [62]. Copyright 2019, *Nature communications*. (**d**) Operational schematic of the proposed nanowire spectrometer [63]. Copyright 2019, *Science*. (**e**) A top view of the designed spectrometer in a false-color scanning electron microscope (SEM) image [64]. Copyright 2020, *Nano letter*. (**f**) Schematic of the monolithic spectrometer [65]. Copyright 2020, *Advanced Optical Materials*. (**g**) Architecture of the single-photon spectrometer [66]. Copyright 2020, *Physical Review Applied*. (**h**) Working principles diagram of the superconducting nanowire single-detector spectrometer. On the right is an SEM of the SNSPD, which meanders in perpendicular directions to reduce photon polarization sensitivity (painted with two different colors) [67]. Copyright 2021, *Nano letter*.

**Figure 4 materials-16-02253-f004:**
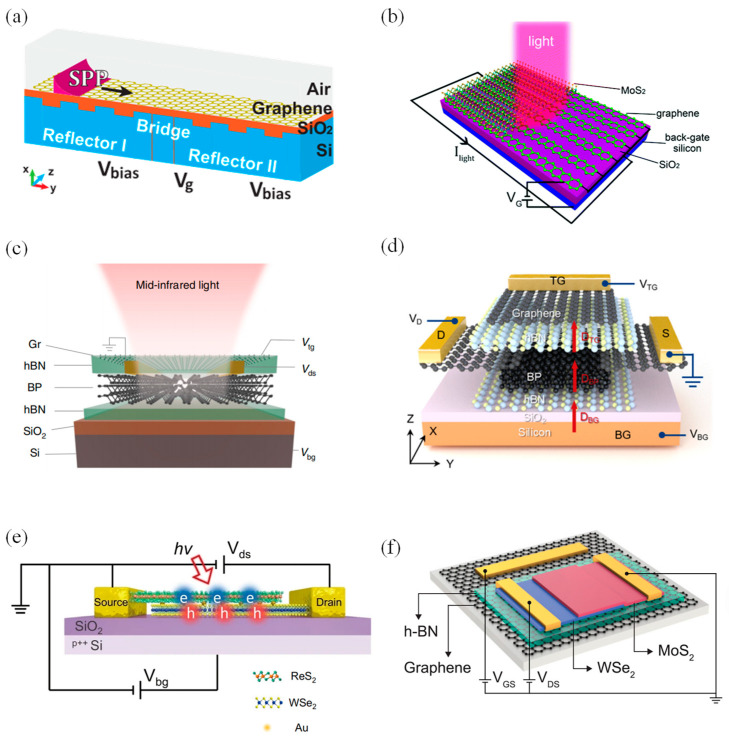
Architecture and schematic diagram of 2D nanomaterials-based micro spectrometers. (**a**) Schematic diagram of the graphene-based Fabry–Pérot spectrometer. Surface plasmon polaritons (SPPs) in graphene propagate across two Bragg reflectors and a bridge between them that are modulated by different voltages. The arrow at the top of the graphene layer indicates the direction of SPPs propagation [78]. Copyright 2016, *Scientific Reports*. (**b**) Schematic of the graphene ribbons and MoS_2_ vertical heterostructure photodetector [79]. Copyright 2018, *Nanoscale*. (**c**) Schematic of the BP spectrometer. Gr, graphene [80]. Copyright 2021, *Nature Photonics*. (**d**) Schematic diagram of a typical dual-gate BP transistor. The applied top and bottom gate voltages (*V*_TG_ and *V*_BG_, respectively) can control the carrier density and the electric displacement field in the sample [81]. Copyright 2022, *Applied Physics Letters*. (**e**) Schematic view of the 2D-vdWH spectrometer. *V*_ds_ and *V*_bg_ are bias voltage and back gate voltage, respectively. ‘*hv*’ and the red arrow indicate the incident light [82]. Copyright 2022, *Nature communications*. (**f**) Schematic of the MoS_2_/WSe_2_ heterojunction spectrometer [83]. Copyright 2022, *Science*.

**Table 1 materials-16-02253-t001:** Progress in micro spectrometers based on nanomaterials.

Year	Nanomaterials Used	Spectral Range	Spectral Resolution	Footprint	Ref.
1997	InAs/GaAs QDs	-	-	-	[42]
2015	InAs QDs	500–1000 nm	-	-	[43]
2018	InAs QDs	500–930 nm	-	-	[44]
2015	CdS/CdSe CQDs	390–690 nm	~3.2 nm	8.5 × 6.8 mm^2^	[45]
2019	HgTe CQDs	1.53–2.08 μm	~180	-	[50]
2020	Perovskite CQDs	250–1000 nm	~1.6 nm	7 × 7 cm^2^	[46]
2021	PbS/PbSe CQDs	900–1700 nm	~6 nm	55 × 55 × 82 mm^3^	[47]
2021	PbS CQDs	400–1600 nm	~40 nm	-	[48]
2022	Perovskite	350–750 nm	~5 nm	440 × 440 μm^2^	[51]
2022	HgTe CQDs	SWIR (Max. 2 μm)	~50 cm^−1^	100 × 100 × 100 μm^3^	[49]
2011	epi-NbN nanowires	VIS-Mid-IR	~170 nm	6 × 8 mm^2^	[59]
2017	GaP nanowires	~610–670 nm	-	Thickness: Micron-scale	[60]
2019	Pt nanowires	~600–800 nm	-	Length: 200 μm	[61]
2019	NbN nanowires	600–2000 nm	~7 nm	12 × 20 mm^2^	[62]
2019	Single CdS_x_Se_1−x_ nanowire	500–630 nm	~10 nm	75 × 0.5 μm^2^	[63]
2019	Multiple Silicon nanowires	450–800 nm	~6 nm	2 × 2 mm^2^	[68]
2020	Superconducting nanowire	800–1550 nm	~30 pm /4 nm	Micron-scale	[64]
2020	CdS_x_Se_1−x_ nanowire	~560–620 nm	~5 nm	Length: 50 μm	[65]
2020	Superconducting nanowire	-	~1 nm	Height: 220 nm	[66]
2021	Superconducting nanowire	660–1900 nm	~6 nm	6 × 6 μm^2^	[67]
2023	Superconducting nanowire	1350–1629 nm	~2 nm	-	[69]
2016	Graphene	Mid-IR	-	Micron-scale	[78]
2018	Graphene ribbons/MoS_2_	6–16 μm	-	-	[79]
2021	Black phosphorus	2–9 μm	~90 nm	9 × 16 μm^2^	[80]
2022	Black phosphorus	Mid-IR	2 cm^−1^	-	[81]
2022	ReS_2_, WSe_2_	1.15–1.47 μm	~20 nm	6 × 4 μm^2^	[82]
2022	MoS_2_, WSe_2_	405–845 nm	~3 nm	22 × 8 μm^2^	[83]

## Data Availability

Not applicable.

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
