# Peer review of "Micro Spectrometers Based on Materials Nanoarchitectonics"

_materials, 2023, doi:10.3390/ma16062253_

Round 1

Reviewer 1 Report

In this review the Authors discuss the research progress of micro spectrometers based on nanomaterials. They also analyze the main limitations and perspectives on micro spectrometers. The manuscript is complete and well written. The explanations are very clear and the discussions are quite convincing. Thus the review paper is very likely to appeal to a broad audience, especially readers interested in the following topics: micro spectrometers, nanomaterials, spectral analysis. The importance of the present review manuscript lies also in the consideration that spectral analysis is an important tool, nowadays widely used not merely in scientific research but also in industry. Since their emerging in the 1990s, micro spectrometers have developed, also thanks to the appearing of nanoscience and nanotechnology, thus relying upon innovative nanomaterials for their design, allowing for a significant reduction of the  spectrometers' size. This aspect, i.e. miniaturization, enabling also portability of the micro spectrometers are of the uttermost importance for certain applications, e.g. on-site detection and real-time monitoring. Hence, I recommend publishing the manuscript as it stands.

Author Response

Please refer to the attachment for specific reply.

Reviewer 2 Report

The authors have done a good job in summarizing the recent achievements on microspectrometers bases nanoparticles, nanowires and 2D-systems. Relevant in the field, necessary for a fast growing field and pertinent for the contents of the special issue. In my opinion the contents of the manuscript fits very well in a special issue devoted to “Colloidal Quantum Dots for Nanophotonics Devices”.

I know that it is very difficult to be aware of all previous works in the field, however I miss a reference to an excellent recent review, namely Li et al., Light: Science & Applications (2022) 11:174,  https://doi.org/10.1038/s41377-022-00853-1.

What does it add to the subject area compared with other published material?
Exhaustive review on recent relevant results on micros-spectrometers based on nanoparticles, nanowires and 2D-Systems

What specific improvements should the authors consider regarding the methodology? What further controls should be considered?
No specific improvement is needed.

The opinion of a second reviewer will be the only further control to be considered.

The conclusions are well supported. And at the end of the paper, the advantages and drawbacks of the considered micro-spectrometers are analyzed as well as the aspects that need further improvement.

 My last comment refers to the quality of the figures (maybe it is only for this provisional version), but sometimes I found it difficult to read the small text in some of the large figures.

Author Response

(The authors gave the same response as above.)

Reviewer 3 Report

This review manuscript actually summarizes interesting subjects. Interesting facts are well included. This review is valuable of being published in well-authorized journals such as Materials. I may recommend publication of this manuscript in Materials. However, some points (especially figure quality) have to be improved by revisions. Please see below.

1) The current title is not so impressive because it is made of rather common words. It is better to add a new conceptual term to add innovative impression. Important factor of this review is not only nanomaterials. Structural controls of nanomaterials are also important. Therefore, I may suggest use of an emerging conceptual term, nanoarchitectonics, in the title (as post-nanotechnology concept, see https://pubs.rsc.org/en/content/articlelanding/2021/nh/d0nh00680g). For example, the title like ... Micro Spectrometers Based on Materials Nanoarchitectonics ... may sound more innovative.

2) Please entirely improve figure quality (resolution and word size).

3) It is better to make one good figure to explain guidance of this review. Probably, modification of Figure 1 may be a good idea. The current Figure 1 is rather simple assembly of these items. If relation of these items are clearly represented in this figure, this figure would become a good guide figure for this review.

4) Conclusions are well written, If one figure to represent these contents of conclusion section, this review can have good ending with better understanding.

Author Response

(The authors gave the same response as above.)

Reviewer 4 Report

The manuscript ID materials-2258079 presents a review related to the assistance of nanomaterials to spectrophotometry instruments. The analysis requires an edition in order to clarify some issues. Please see below some points to the authors:

1.   The title and keywords should be more specific. And in addition, the fonts within the text in most of the figures should be enlarged.

2.   From the first paragraph of the introduction is still unclear the area of the report. There are many types of spectrometers: Mass, Raman, Infrared, etc. Please edit.

3.   If the authors are describing UV-vis spectra in the introduction, the expression “Every substance shows its specific spectrum, and the spectrum is thus regarded as the fingerprint that identifies matter” should be edited, since the same optical spectra can be associated to different substances.

4.   Several optical properties like irradiance and influence of polarization in the different spectrometers should be described with better details.

5.   A confrontation of the advantages and disadvantages of the different types of spectrometers classified should be discussed.

6.   Table 1 is useful, but in order to support Figure 1, a roadmap describing the representative references and date of the publications associated to structures and algorithms of these micro spectrometers based on nanomaterials is suggested to be included.

7.   The authors should state within the abstract or introduction what this report adds to literature in respect to other comparative reports in the topic.

8.   It would be interesting if the authors could extend some comments for the assistance of micro spectrometers in fluid detection. You can consider for instance studies involving nanoparticles and their optical properties in different liquids: https://doi.org/10.1016/j.ijleo.2019.01.042

9.   In my opinion, the number of references for a review in this prestigious journal is typically larger. The authors could update to the period of this review stage the bibliography in the topic. You can see for instance: https://opg.optica.org/ol/abstract.cfm?URI=ol-47-11-2923

10.               Regarding that this is a review of selected literature, it is suggested to split the collective citations in individual form with expressions that justify each selected reference to be mentioned in this work.

Author Response

(The authors gave the same response as above.)

Round 2

Reviewer 4 Report

The authors have successfully addresed most of the points raised in the review stage. The analysis is interesting and the conclusions are useful in this topic of nanocale spectrophotometers. Then, in my opinion, this work can be considered for publication in present form.